# Individual differences in executive functions and theory of mind mediate the relation between academic skills from kindergarten to 5th grade

Sarah Le Diagon [1]*, Marie Jacquel[1,2], Jean-Baptiste Van der Henst[1], Jérôme Prado[1]*

**1** Centre de Recherche en Neurosciences de Lyon (CRNL), INSERM U1028 - CNRS UMR5292, Université de Lyon, Lyon, France, **2** Université Paris Cité, LaPsyDÉ, CNRS, Paris, France

* sarahlediagon@gmail.com (SLD) or jerome.prado@cnrs.fr (JP)

## Abstract

Individual differences in early academic skills at school entry are known to predict later academic outcomes, as demonstrated primarily by studies conducted in the United States (US). However, the mechanisms underlying this association remain unclear. In a country where early childhood education is more homogeneous than in the US (i.e., France), this study examined the strength of that predictive relationship and explored whether it could be partly explained by two domain-general mechanisms that are fundamental to effective learning in a school context: executive functions and children's ability to navigate social relationships. We measured math and reading skills in a cohort of 95 French children in both kindergarten and 5th grade, while also assessing their self-regulation, working memory, planning, theory of mind, and social behaviors at one or both time points. Results confirmed within- and cross-domain associations between early and later academic skills that were comparable to those found in previous studies. Self-regulation, working memory, and theory of mind all mediated both within- and cross-domain relationships. However, these mediations were systematically partial, meaning that early measures of academic achievement remained particularly strong predictors of later academic success. These findings suggest that domain-general cognitive processes, such as executive functions and social cognition, may play a role in explaining the relation between early and later academic achievement. However, a significant part of that relation may still be explained by domain-specific skill-building mechanisms.

## Introduction

School achievement levels in reading and math are predictive of fundamental adult outcomes, including socio-economic status [1], employability [2], salary [3] and physical and mental health [4]. Therefore, the wide inequalities in literacy and numeracy performance that are often observed between individuals at school exit have

**Data availability statement:** All data files are available from the OSF database (https://osf.io/3rmc9).

**Funding:** JP 00123415 / WB-2021-38649 Fondation de France https://www.fondationde-france.org/fr/ The funder did not play any role.

**Competing interests:** The authors have declared that no competing interests exist.

raised concerns among education experts and policymakers [5]. A substantial body of longitudinal evidence indicates that later individual differences in academic performance are typically predicted by performance variability at school entry. For example, studies—mostly conducted in the United States—have shown that children's literacy and numeracy skills in kindergarten are predictive of their later literacy and numeracy skills in elementary school [6–9], middle school [10] and even high school [11]. While this has led to increased interest in educational programs aimed at improving early math and reading achievement levels, the success of these programs hinges on understanding the mechanisms that underlie the relation between early and later achievement. These mechanisms, however, remain poorly understood.

### The "skill-building" hypothesis

The predominant hypothesis for explaining the relation between early and later academic achievement is that of "skill-building" [12]. The idea is that early competences may provide the foundations upon which future and more complex skills are built. Skill-building would be particularly important for skills that are cumulative, such as reading and math. In the reading domain, for example, mastering letter-sound associations may allow children to recognize words, which may in turn support the development of vocabulary and help reading comprehension [13]. In the math domain, the early understanding of counting principles and cardinality may provide the foundation for arithmetic, which may itself allow for the acquisition of algebra [14]. Overall, the skill-building hypothesis is supported by longitudinal studies showing that early foundational competences do predict later literacy and numeracy skills. For example, phonological awareness, print knowledge and oral language (i.e., vocabulary and oral comprehension) are predictive of later reading skills [15–20]. Number knowledge, symbolic numerical skills, and basic arithmetic skills are also predictive of later math achievement [8,21–23].

However, some have cast doubt on skill-building as a unique mechanism explaining the relation between early and later skills for three main reasons. First, even though they may control for a range of variables, longitudinal studies that rely on correlational evidence cannot be comprehensive and necessarily omit measures (e.g., behavioral, environmental, or cognitive) that may better account for the relation between early and later achievement. Therefore, Bailey et al. (2018) argued that such studies do not constitute "sufficiently risky tests" for the skill-building hypothesis, as correlations between early and later skills may exist even in the absence of skill-building [24]. Second, interventional programs aimed at improving children's emergent academic skills show mixed results. Though these interventions are typically able to improve literacy and numeracy skills of young children in the short-term, the effect frequently decreases or even disappears several months or years after the interventions [24–32]. This phenomenon—commonly referred to as "fade-out" in the literature [33,34]—is inconsistent with the skill-building hypothesis, which would instead predict a greater or at least a sustained effect in the long term [34]. Third, longitudinal studies have shown cross-domain relations between early and later academic skills. For example, early math skills have been found to predict later

reading skills [6,7,10], while early reading skills have been found to predict later math skills [7]. This suggests that the mechanisms underlying the relation between early and later academic competences might be more domain-general (i.e., not specific to learning math or reading) than assumed by the skill-building hypothesis. For instance, these mechanisms could stem from stable environmental or individual factors that may generally influence learning throughout schooling [24]. Below we focus on two of these potential domain-general mechanisms, i.e., executive functions as well as social cognition and behavior.

## Executive functions

One potential mechanism that might underlie the relation between early and later academic achievement is children's level of executive functioning, which typically includes working memory, inhibition, flexibility, behavioral self-regulation, and planning [35–37]. Indeed, studies have shown either cross-sectional associations or short-term longitudinal relations between measures of executive functions and success in reading and math in both kindergarten [38–40] and elementary school [41–43]. Notably, studies suggest that there is a bidirectional relation between academic skills and executive functions. On the one hand, executive functions may influence later academic skills because they may allow children to consistently adapt to changing classroom environments and may help them acquiring more complex skills [44–46]. The idea that executive functions influence academic learning is supported by two lines of evidence. First, longitudinal studies have demonstrated that early executive skills in children can predict their later success in reading and math [47–51]. Second, studies have also shown that interventions aimed at improving executive functions lead to improved academic skills, suggesting a causal relation [52–55]. Note, however, that this finding is relatively controversial, as a number of other intervention studies have failed to find that enhancing executive functions necessarily lead to better academic skills [56–60].

On the other hand, the acquisition of early reading and math may also help scaffold the development of children's executive functions. This is because children with higher levels of reading or math skills may face more complex learning situations in these domains [61]. These children may naturally engage their executive functions in increasingly complex tasks, which would then promote the growth of executive skills [62]. For example, reading activities that require holding and manipulating information, or the rapid and recurrent shift from phonology to semantics during text reading, are likely to engage working memory [63] and flexibility [64]. As another example, math activities such as coordinating and maintaining operations during mental calculation and problem solving may recruit attentional control and working memory [65,66]. Consistent with this hypothesis, longitudinal studies have shown that early literacy and math skills may predict future executive skills several years later [47,67,68]. Some interventional studies even show that improving academic competences may improve executive functions, suggesting far-transfer [58,61,69–71]. However, this finding is debated in the literature as other studies fail to find such transfer effects [52,72]. Nonetheless, taken together, previous studies suggest that individual differences in executive functioning are both influenced by academic skills and also influence the acquisition of new skills. Therefore, executive functions are a good candidate mechanism for explaining the relation between early and later academic skills.

## Social cognition and social behavior

Critically, schools are also social environments. As such, learning at school is fundamentally a social process [73–77] that involves managing relationships with teachers and peers. Therefore, there may be a relation between academic skills and children's abilities to navigate the social world. These abilities may encompass social cognition skills, which center on the ability to understand our own and others' mental states (e.g., thoughts, intentions, emotions, desires, or beliefs) but also include the ability to perceive social relationships (alliance, friendship, dominance) and form beliefs associated with social categories or gender (e.g., stereotypes). These abilities may also include social behaviors, which describe the behaviors that are needed to develop socially appropriate responses and build positive relationships with others (e.g., sharing, communicating, cooperating, and knowing how to solve social problems). The idea that academic achievement relates to social cognition and behavior is supported by a number of cross-sectional and short-term longitudinal studies.

For example, proficiency in Theory of Mind (ToM)—the ability to infer mental states—is linked to greater academic skills (see [78] for a meta-analysis) both in preschool [38,79,80] and in elementary school [81–85]. Other studies have found a relation between children's socio-emotional skills (e.g., emotion recognition and emotion understanding) and math and reading skills in preschool [86–89] and in elementary school [90–92]. Social skills (e.g., cooperating, communicating, and sharing with others, being empathetic, taking responsibility, accepting others) have also been found to be associated with concurrent levels of academic achievement [93–96].

Interestingly, the relation between academic achievement and social cognition and behavior is likely to be bidirectional. On the one hand, academic achievement may enhance social cognition and behavior. For instance, as children learn to read, they may become more proficient at comprehending and interpreting social narratives, which could improve their understanding of others' perspectives and emotions [97,98]. Developing academic skills might also boost children's confidence and self-efficacy [99,100], which might in turn facilitate better interactions with peers and teachers. In keeping with the idea that academic skills may enhance social cognition and behavior, longitudinal evidence indicates that improved linguistic skills in kindergarten predict enhanced theory of mind skills in elementary school [81,101]. Similarly, better math and reading skills in kindergarten predict better social behavior in middle school [102].

On the other hand, children who are better able to navigate the social world are also more likely to establish better social relationships, which might promote more effective collaborative learning and in turn better academic skills. Enhanced levels of ToM might also be helpful for understanding implicit language (e.g., irony, metaphors) [103], which may in turn improve reading comprehension. Therefore, social cognition and behavior may influence academic achievement. This is, for example, suggested by longitudinal evidence showing that late academic skills are predicted by earlier capacities of theory of mind [81,104–106], socio-emotional processing [107–110], and social skills (e.g., aggressive and prosocial behavior, interpersonal skills) [102,111–113]. Interventional studies also suggest that improving socio-emotional skills may increase academic achievement [114–116]. Overall, then, academic skills may affect social cognition and behavior as much as the other way around. This makes social cognition (notably ToM) and social behavior interesting candidate mechanisms for explaining the relation between early and later academic skills.

## The current study

The goal of the present study was twofold. First, the large majority of studies that have observed a relation between early and later academic achievement have been conducted in the United States. Because compulsory schooling in the United States starts in kindergarten (i.e., age 5), there is a large variability in children's skills at school entry, which might exacerbate the relation between kindergarten skills and later skills [117]. Therefore, a first aim of the present study was to test whether there would be similar within- and cross-domain longitudinal relations between reading and math skills from kindergarten to the end of elementary school (i.e., 5th grade) in a country where preschool education is homogenous and compulsory from age 3, i.e., France. To our knowledge, only one previous study [15] has evaluated the relation between emerging literacy skills in kindergarten and reading skills at the end of elementary school in France. However, that study did not include any measure of numeracy skills, preventing the authors to evaluate the existence of a relation between early and later math skills, as well as a cross-domain relation between math and reading. Here, we measured both math and reading skills in a longitudinal sample of children from kindergarten to 5th grade. We hypothesize that a greater homogeneity in the early childhood education system in France would reduce interindividual differences in academic achievement at elementary school entry. This should result in a weaker relationship between early and later academic skills.

Second, should a relation be observed between academic achievement in kindergarten and 5th grade, another goal of the study was to test whether that relation might be explained by individual differences in either executive functions or social cognition and behavior. Therefore, in addition to measures of math and reading skills, we gathered in the same longitudinal sample measures of executive functions (i.e., working memory, behavioral self-regulation, and planning), social cognition (i.e., ToM), and social behavior (i.e., sharing, distributive justice, and social problem-solving)

while children were in kindergarten, 5th grade, or both. We then tested whether those measures mediated any potential within- and cross-domain relation between early and later academic skills. To our knowledge, no previous study has investigated the mediating role of ToM and social behavior in the relation between early and later academic skills, either in isolation or with executive functions. However, two studies have investigated whether executive functions might mediate that relation from early grades to either the end of elementary school [68] or high school [118]. The results were inconsistent. On the one hand, Watts et al. (2015) failed to find that executive functions mediated the relation between early and later math skills in the United States. On the other hand, ten Braak et al. (2022) did observe that executive functions mediate the effect of kindergarten math on 5th grade math and reading in Norway. However, that latter study used a single measure of behavioral self-regulation that did not make it possible to assess the role of other executive functions. They also explored whether executive functions mediated the relation between early math and later reading but not the other way around. In the present study, a wider range of executive measures were collected in addition to the self-regulation measure also used in ten Braak et al. (2022). We also collected reading and math measures both in kindergarten and in 5th grade, allowing an exploration of all possible cross-domain relations. Given the associations between academic skills and executive functions, social skills, as well as social cognition, we hypothesize that individual differences in all of these skills would mediate the relation between early and later academic abilities. The hypotheses, methods, materials, and planned analyses of the present study were pre-registered (https://osf.io/8f6tb) https://osf.io/8f6tb/?view_only=a0988626c2784a56955a602f2de5563e.

Overall, the aim of this work was to assess the extent to which longitudinal relations between early and later academic skills are present within the French school context and to better understand the underlying mechanisms of these relations. This objective was achieved.

## Methods

### Participants

The present study is a longitudinal follow-up, in 5th grade, of children whose academic skills, executive functions, ToM skills, and social behavior were assessed while they were in kindergarten. A previous analysis of these data with respect to the type of pedagogy attended by children in kindergarten is reported in a previous report [99]. The original sample included 131 children from three cohorts, who were tested at the end of their kindergarten year (June 2017, June 2018, and June 2019). All children attended the same school, located in the Lyon suburban area in France. Yearly household income, estimated from a subsample of parents (n = 51), ranged from less than €18,000 to €75,000, with an average of €27,000. Given that the French median household income was of €30,260 in 2018 (https://www.insee.fr/fr/statistiques/5371205?sommaire=5371304), family socio-economic status ranged from very low to very high but was relatively low on average. Data on participant ethnicity were not collected because the collection of such data is in principle illegal in France and would require an exceptional waiver from government agencies that we did not seek. Five years later, we were able to locate 105 of the children from the original sample, who either still attended the same school (n = 76) or attended another school in the same neighborhood (n = 29). For this second data collection, children were tested at the end of 5th grade (March 2022, March 2023, and March 2024). The study received ethical approval from the INSERM institutional review board (22–881). Parents gave their written informed consent and children gave their assent to participate.

Among the children we were able to locate, 10 were excluded from the analyses because we were not able to secure parental consent (n = 4), because they were outlier on several measures (n = 1), or because they had skipped a grade or repeated a year between kindergarten and 5th grade (n = 5). Therefore, the final sample included 95 children (43 girls) followed from kindergarten ($M_{age}$ = 5.99 years, SD = 0.28) to 5th grade ($M_{age}$ = 10.74 years, SD = 0.28). From 1st to 5th grade, all children followed the exact same conventional French public-school curriculum (https://www.education.gouv.fr/programmes-et-horaires-l-ecole-elementaire-9011).

## Preregistration

The material, hypotheses and analysis plan were pre-registered on the Open Science Framework (OSF) at https://osf.io/8f6tb https://osf.io/8f6tb/?view_only=a0988626c2784a56955a602f2de5563e. Four main changes were made to the preregistration. First, we slightly modified the inclusion criteria. Though we omitted this criterion in the preregistration, we chose to exclude from our analyses children who had repeated or skipped a grade, since their developmental stage is not the same as that of other children. We also chose to not exclude one child whose parent was a kindergarten teacher, as this was unlikely to affect longitudinal analyses. Finally, we did not exclude participants with potential learning disabilities because we did not have sufficient information to characterize these disabilities. Second, given that our main analyses center on mediations, we focused on frequentist rather than Bayesian analyses in the present report. Third, we did not investigate the relation between academic progression from age 3 to age 5 and academic achievement in 5th grade because the sample size was too limited. Fourth, although we had planned to use overall scores (from 0 to 18) on the Story-Based Empathy Task in 5th grade, we realized that this was not appropriate as it included a measure of causal reasoning that did not involve any attribution of intention and emotion. Therefore, we removed this measure from our calculation of ToM scores on this task. Because ToM scores from that task were correlated with scores on the other measure of ToM (i.e., Reading the Mind in the Eyes) in 5th grade (r(87) =.30, p = .005), we combined these measures to form a composite score of ToM in 5th grade for parsimony (see below).

## Justification of sample size

The present sample of children was constrained by the initial data collection in kindergarten. We expected in our preregistration a final sample of n = 118. Though attrition was higher than expected, our final sample size of n = 95 still allows us to detect a minimum effect size of $f2 = .08$ with a power of 80% ($\alpha = 0.05$), in a multiple regression with 5 predictors. This is smaller than the effect sizes reported in previous studies linking early and late skills (typically reporting large effects beyond $f2 = .15$ [7,11]). Thus, our study should be sufficiently powered to detect the anticipated relations.

## Materials

Because the original kindergarten sample was also part of another project [99], early assessments included a wide range of literacy (i.e., decoding, vocabulary, phonological awareness, and pragmatic comprehension in kindergarten) and numeracy measures (i.e., math problem solving, basic quantitative and counting knowledge). Literacy and numeracy measures in 5th grade included decoding skills for reading, as well as math problem solving and arithmetic fluency for math. To accurately assess the relation between early and later academic achievement, we selected the two measures that targeted the exact same construct in kindergarten and 5th grade, i.e., decoding skills for reading and math problem solving for math. The tests used to measure these skills are described in Table 1. Other literacy and numeracy measures are described for completeness in Table S1.

Children were also tested on a range of executive functions in both kindergarten and 5th grade, as described in Table 2. These included an assessment of working memory skills, using the same instrument at both time points (i.e., the Corsi Block Tapping task). Note that a measure of short-term memory was also collected as part of the Corsi Block Tapping task at both time points. Though short-term memory is typically not considered an executive function per se, we still chose to analyze these scores as it may be informative as a control measure regarding the specificity of any potential relation between working memory skills and academic achievement. Other tests included measures of behavioral self-regulation and planning, exclusively collected in kindergarten (see Table 2).

Finally, ToM and a range of tasks assessing social behavior were also presented to children in both kindergarten and 5th grade, using age-appropriate instruments (see Table 3). ToM was notably assessed in 5th grade through a composite measure of both empathy and emotion recognition. However, we also report correlations between each measure taken separately and academic outcomes to assess the specificity of measures of empathy and emotion recognition. Most tests

**Table 1. Descriptions of the Tests Measuring Academic Skills by Grade.**

| Skill | Test | Description | Time of testing | | Scoring |
|---|---|---|---|---|---|
| | | | K | 5th grade | |
| Reading (Decoding) | Reading subtest of the Evaluation Des fonctions cognitives et des Apprentissages (EDA) [119] | Decode letters, then digraphs, increasingly difficult words, and finally sentences. The test was stopped if a child was not able to read any of the letters or any of the digraphs. | X | | Number of correctly completed items (0–70) |
| | Reading fluency subtests of the Evaluation du langage oral et écrit 6–12 (EVALEO) [120] | Read out loud two 450-word texts as quickly and as accurately as possible within 2 minutes. One text was sensical while the other was nonsensical. | | X | Number of words correctly read on average for both texts (0–450). |
| Math (Problem solving) | Applied Problems subtest of the Woodcock-Johnson III [121]. | Respond to 63 increasingly difficult math problems. The first items involved counting, simple subtraction and addition, clock reading and calculating with coins. Items then progress onto word problems. The test is stopped after participants made six consecutive errors.[1] | X | X | For each time point, number of correctly completed items (0–63). |

*Notes.* All tests were administered individually.

[1]The test was administered from item 15 onwards in 5th graders.

**Table 2. Descriptions of Tests Measuring Executive Functions by Grade.**

| Skill | Test | Description | Time of testing | | Scoring |
|---|---|---|---|---|---|
| | | | K | 5th grade | |
| Short-term[2] and Working memory | Corsi Block Tapping task [122] | Repeat a spatial sequence shown by the experimenter by touching blocks glued onto a wooden board. The sequence progressively increases in difficulty. The sequence is first repeated in the same order as the experimenter (up to 9 blocks) to measure short-term memory. It is afterwards repeated in the reverse order (up to 6 blocks in kindergarten and 9 blocks in 5th grade[1]) to measure working memory. For each sequence, there are four chances to succeed. | X | X | For each time point, maximum number of blocks correctly repeated forwards (score = 0–9) for short-term memory and maximum number of blocks correctly repeated backward (score = 0–6) for working memory. A composite score was calculated across time points for short-term and working memory. |
| Self-regulation | Head Toes Knees Shoulders (HTKS) task [123] | Perform a gesture that is opposite to the one shown by the experimenter (e.g., touch the toes when asked to touch the head and the other way around). The test is made progressively more difficult by adding knees and shoulders commands. Children get two points every time they directly performed the correct action, one point if they have to self-correct their action and no point if they perform an incorrect action. | X | | Sum of points (0–52). |
| Planning | Planification subtest of the EDA [119] | Complete three mazes of progressive difficulty. Each completed maze is worth 10 points. Children have a maximum of 120 seconds to complete each maze and lose one or two points depending on their completion time. The test is stopped if a child scored 0 at a maze. | X | | Sum of the points for all the mazes completed (0–30). |

*Notes.* All tests were administered individually.

[1]With the exception of one cohort of 5th graders for which the sequence was presented up to 6 blocks.

[2]Short-term memory skills were used as a control measure, as they are typically not considered an executive function per se.

**Table 3. Descriptions of Tests Measuring Social Cognition and Behavior by Grade.**

| Skill | Test | Description | Time of testing | | Scoring |
|---|---|---|---|---|---|
| | | | **K** | **5th grade** | |
| ToM | Wellman & Liu task [126] [1] | Understand the mental state of a character. The task involves 5 stories measuring the complexity of theory of mind understanding (diverse desires, diverse beliefs, diverse knowledge, false belief, hidden emotion). The test is stopped after two failed stories. | X | | Number of stories completed successfully (0–5). |
| | Story-based Empathy Task (SET) [127] [2] | Complete a 3-panel comic strip with a fourth panel out of three possibilities. Three different categories of comic strips were presented: stories involving the identification of intentions (6 items), stories involving the identification of emotional states (6 items), and stories that do not involve identification of either intentions or emotions (filler items, 6 items). | | X | Number of correctly completed stories involving intentions or emotional states (0–12). That score was combined with the score from the Reading the Mind in the Eyes to obtain a composite score of ToM skills in 5th grade. |
| | Reading the Mind in the Eyes (RME) task [128] [2] translated in French [129] | Select out of four propositions the word that best describes what a person in a photograph is feeling or thinking. Photographs were black-and-white and depicted the eye region of a different individual (28 items). | | X | Number of correctly recognized emotion or thought (0–28). That score was combined with the score from the Story-based Empathy Task to obtain a composite score of ToM skills in 5th grade. |
| Sharing | Dictator Game [130] [1] | Share stickers [3] with other children. Children choose their 10 favorite stickers [3] among a choice of 30. They are then told that not enough stickers [3] are available for other children. Participants could donate any number of their stickers [3] to other children (the choice is made while the experimenter is looking away). | X | X | Number of stickers [3] given (0–10). A composite score was calculated across time points. |
| Distributive justice | Resource allocation task (inspired by [131]) [1] | Distribute candies to characters in situations of inequality. Three situations are presented (resource inequality between a rich and a poor character, unequal contribution to a common good between a hard-working and a lazy character, and a power inequality between a dominant and a subordinate character). After each situation, children are told that both characters love candies. They have to distribute 4 candies to the characters the way they want. | X | | Number of candies allocated to the disadvantaged characters in each of the three conditions (0–4) and in total (0–12) |
| Social problem solving | Object acquisition item of the Social Problem Solving Task-Revised (SPST) [132] [1] | Solve a problem involving a social component. Two cartoon children are shown, one reading a book and the other standing behind her/him (the gender matched the gender of the participants). Children are told that the character reading the book has been reading it for a very long time and that the other character wants to have the book. Participants must imagine what the character could say or do to get the book, what else the character could say or do to get the book, and what they could themselves say or do if they want a book that another child is already using. | X | | Number of responses with reference(s) to justice/sharing (0–3) [133], as well as flexibility (i.e., level of novelty) of the second response in relation to the first response (0–3) [134]. |

*Notes.*

[1]Test administered individually.

[2]Test administered in a whole-class setting.

[3]Stickers were replaced with candies in 5th grade.

of social behavior were only used in kindergarten, with the exception of the Dictator game, which was used to measure sharing behavior in both kindergarten and 5th grade.

For working memory and sharing behavior, we created composite scores across time points because: (1) identical instruments were used at both time points (Corsi Block Tapping task for working memory measures and Dictator Game for sharing behavior), (2) these measures showed significant correlations between kindergarten and 5th grade

assessments (r = .29, p < .005 for working memory; r = .34, p < .001 for sharing behavior), and (3) these constructs represent relatively stable individual traits that have demonstrated moderate stability across development in previous longitudinal research [124,125]. Therefore, composite scores represent the general capacity of children in working memory and sharing behavior across the studied developmental period, allowing us to examine how these traits might mediate the relation between early and later academic skills (note that we also created a composite measure of short-term memory so that it was comparable to the composite measure of working memory). Nonetheless, we also conduct separate mediation analyses using separate time point measures rather than composites to examine the sensitivity of our findings.

## Procedure

In kindergarten, children were systematically tested individually in a quiet room of their school. The tests were administered by different experimenters (graduate student, research assistants and undergraduate students) in five sessions of approximately 15–20 minutes. Order of sessions was randomized on a child-to-child basis. In 5th grade, children were tested individually in a quiet room of their school for one hour and collectively with the whole class for another hour (see Tables 1–3). No feedback was given to children during testing.

## Data analysis

Data analysis was performed with the Jamovi software (version 2.3.28) and R (version 4.3.0) with Rstudio. Raw scores were converted to Z-scores to facilitate comparison of regression coefficients and calculation of composite scores across time points whenever the same instrument was used in kindergarten and 5th grade (i.e., Corsi and Dictator game) or across instruments whenever the same construct was measured at a single time point (i.e., SET and RME in 5th grade).

To assess the within and cross-domain longitudinal relations between early and later academic skills, multiple regression analyses were performed, controlling for age and sex. In these regression analyses, academic skill in kindergarten (reading or math) was systematically considered the independent variable while academic skill in fifth grade (reading or math) was considered the dependent variable.

We then used mediation analyses to explore whether executive functions, ToM, or social behavior explained the relation between early and later academic skills, systematically controlling for age and gender. A mediation analysis aims to test whether the relation between an independent variable and a dependent variable is explained by a third variable, known as the mediator. This analysis typically involves estimating three main pathways: (1) Path a (i.e., the effect of the independent variable on the mediator), (2) Path b (i.e., the effect of the mediator on the dependent variable while controlling for the independent variable), and (3) Path a*b (i.e., the indirect effect and formal test of mediation, indicating whether the relation between the independent and the dependent variable, or Path c, is significantly reduced by including the mediator in the model). In our main mediation analyses, Path a reflected whether academic skills in kindergarten predicted executive functions, ToM, or social behavior. Path b reflected whether executive functions, ToM, or social behavior predicted academic skills in 5th grade (controlling for academic skills in kindergarten). Finally, path a*b was the formal test of mediation, reflecting whether these effects were jointly strong enough to mediate the relation between early and later academic skills.

Mediation models were only tested when there were significant associations between all three variables of the mediation model (the independent variable, the dependent variable, and the mediator). In such cases, mediation paths were estimated using bias-corrected bootstrapping analyses with 1,000 iterations, with 95% confidence intervals. Bootstrapping was preferred over Sobel test because it is more robust for small to moderate sample size [135,136]. Effects were considered significant when confidence intervals did not include zero.

## Results

Raw scores for all tests and time points are shown in Table S2. Zero-order correlations between all standardized scores are presented in Table 4. In what follows, we first investigate the within and cross-domain relations between early and later academic skills before turning to potential mediators of these relations.

### Within and cross-domain relations between early and later academic skills

As shown on Table 4, both within-domain longitudinal correlations between academic scores in kindergarten and in 5th grade were significant. The correlation was of medium size for reading skills (r(92)=.31, p=.002) and of large size for math skills (r(92)=.52, p<.001). Both cross-domain longitudinal correlations were also significant, with medium correlations between math skills in kindergarten and reading skills in 5th grade (r(92)=.33, p=.001) and between reading skills in kindergarten and math skills in 5th grade (r(92)=.42, p<.001). Within- and cross-domain longitudinal associations were confirmed by multiple regression analyses controlling for age and gender. That is, math scores in kindergarten predicted math scores in 5th grade (β=.53, p≤.001, SE=0.09) and reading skills in kindergarten were also predictive of reading skills in 5th grade (β=.31, p≤.01, SE=0.10). Similarly, math scores in kindergarten predicted reading scores in 5th grade (β=.34, p≤.01, SE=0.11), while reading scores in kindergarten predicted math scores in 5th grade (β=.40, p≤.001, SE=0.09). Therefore, our results largely replicated the finding that earlier academic skills are predictive of later skills, with both within-domain and cross-domain relations.

### The role of executive functions in the relation between early and later academic skills

We then examined whether executive functions mediated the relation between early and later reading and math skills. As shown in Table 4, we did not find any measures of executive functions that were related to reading scores in both kindergarten and 5th grade. However, measures of behavioral self-regulation (as measured by the HTKS in kindergarten), short-term memory, and working memory (as measured by the Corsi task at both time points) were all related to math skills in both kindergarten and 5th grade, with correlations ranging from small to large. Moreover, the measure of self-regulation was related to both reading scores in kindergarten and math scores in 5th grade, with small to medium correlations.

**Table 4. Zero-order Correlations Between all Standardized Variables.**

| Skill | 1 | 2 | 3 | 4 | 5 | 6 | 7 | 8 | 9 | 10 | 11 | 12 | 13 | 14 |
|---|---|---|---|---|---|---|---|---|---|---|---|---|---|---|
| 1. Reading (K) | — | | | | | | | | | | | | | |
| 2. Reading (5th) | **0.31** | — | | | | | | | | | | | | |
| 3. Math (K) | **0.46** | **0.33** | — | | | | | | | | | | | |
| 4. Math (5th) | **0.42** | 0.19 | **0.52** | — | | | | | | | | | | |
| 5. Short-term memory (K and 5th) | **0.21** | 0.18 | **0.27** | **0.29** | — | | | | | | | | | |
| 6. Working memory (K and 5th) | 0.19 | 0.14 | **0.45** | **0.53** | **0.43** | — | | | | | | | | |
| 7. Self-regulation (K) | **0.27** | 0.18 | **0.45** | **0.49** | 0.17 | **0.30** | — | | | | | | | |
| 8. Planning (K) | 0.16 | **0.33** | 0.25 | 0.19 | **0.32** | 0.23 | 0.21 | — | | | | | | |
| 9. ToM (K) | -0.04 | **0.26** | 0.11 | 0.10 | 0.06 | 0.14 | **0.27** | 0.09 | — | | | | | |
| 10. ToM (5th) | **0.21** | **0.37** | **0.35** | **0.37** | 0.11 | **0.26** | **0.26** | **0.37** | 0.18 | — | | | | |
| 11. Sharing (K and 5th) | 0.02 | -0.03 | 0.01 | 0.08 | 0.11 | **0.21** | 0.08 | 0.06 | 0.19 | 0.06 | — | | | |
| 12. Distributive justice (K) | -0.05 | 0.17 | 0.05 | 0.02 | 0.02 | 0.04 | 0.15 | **0.31** | **0.20** | 0.10 | 0.05 | — | | |
| 13. Social problem-solving flexibility (K) | -0.02 | 0.03 | 0.14 | 0.05 | -0.01 | 0.16 | 0.09 | -0.002 | 0.12 | -0.08 | -0.04 | 0.14 | — | |
| 14. Social problem-solving justice (K) | -0.13 | 0.13 | 0.06 | -0.06 | 0.06 | -0.16 | -0.04 | **0.37** | 0.06 | **-0.23** | **-0.24** | 0.07 | 0.13 | — |

*Notes.* Time point of administration is indicated in parentheses. Significant correlations (p<.05) are in bold.

Because these associations satisfied our criterion for a potential mediation (see Methods), we then tested whether self-regulation, short-term memory, or working memory mediated the relation between early math and reading skills and later math skills. Mediation models with path estimates are shown on **Fig 1**. The within-domain relation between math scores in kindergarten and 5th grade was partially mediated by both self-regulation skills (indirect effect = 0.131, 95% CI = [0.03, 0.29], indirect/total effect = 0.25) and working memory skills (indirect effect = 0.151, 95% CI = [0.03, 0.32], indirect/total effect = 0.28). Short-term memory, which is not considered an executive function per se, did not mediate that relation (indirect effect = 0.033, 95% CI = [-0.01, 0.09], indirect/total effect = 0.06). Moreover, the cross-domain relation between reading scores in kindergarten and math scores in 5th grade was also partially mediated by self-regulation skills (indirect effect = 0.094, 95% CI = [0.01, 0.22], indirect/total effect = 0.24). Because working memory and short-term memory measures were calculated using composite scores across time points, we also conducted sensitivity analyses using separate time point measures rather than composites (see Table S3). These analyses revealed that kindergarten working memory significantly mediated the relationship between early and later math skills (indirect effect = 0.17, 95% CI [0.06, 0.33], p = .002), whereas 5th grade working memory showed a smaller and non-significant mediation effect (indirect effect = 0.04, 95% CI [-0.02, 0.14], p = .20). While the point estimates might suggest potentially stronger mediation effects for working memory measure at kindergarten, the overlapping confidence intervals indicate that we cannot conclusively determine that one time point's measurement provides a significantly stronger mediating effect than the other. Neither kindergarten nor 5th grade short-term memory significantly mediated the relationship between early and later math skills. Therefore, our

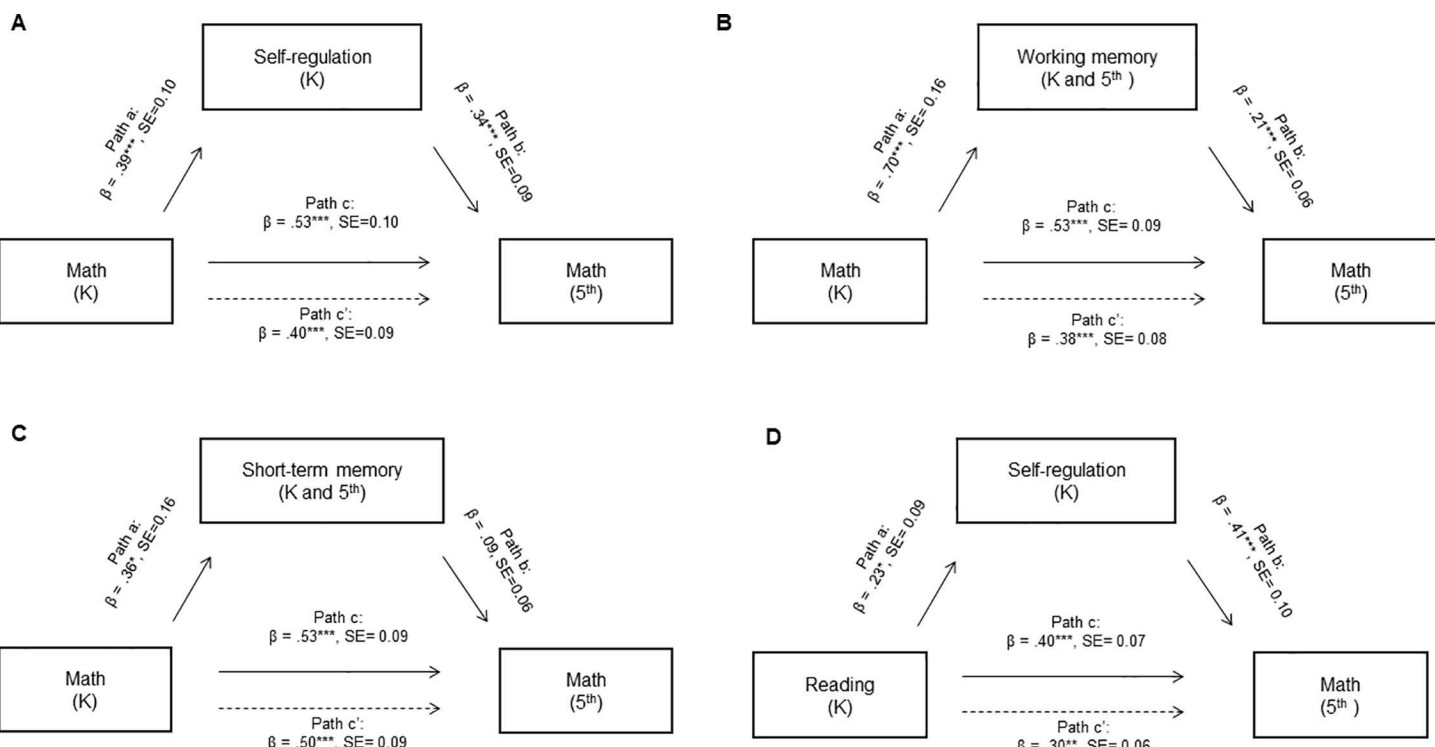

**Fig 1. Mediation Paths for Models of Executive Functions Explaining the Relation Between Early and Later Academic Skills.** (A) Mediation of the relation between early and later math skills by self-regulation measured in kindergarten. (B) Mediation of the relation between early and later math skills by working memory measured in both kindergarten and 5th grade. (C) Mediation of the relation between early and later math skills by short-term memory measured in both kindergarten and 5th grade. (D) Mediation of the relation between early reading skills and later math skills by self-regulation measured in kindergarten. Solid lines between the independent and the dependent variable are total effects, and dashed lines are direct effects. * p < .05, ** p < .01, *** p < .001.

results are compatible with the idea that (1) early math skills may promote behavioral self-regulation and working memory, which in turn may promote later math skills (i.e., a within-domain relation), and that (2) early reading skills may promote behavioral self-regulation, which in turn may promote later math skills (i.e., a cross-domain relation).

**The role of social cognition and behavior in the relation between early and later academic skills**

Finally, we examined whether ToM and social behavior mediated the relation between early and later reading and math skills. We did not find that social behavior was associated with either math or reading skills in both kindergarten and 5th grade. However, ToM skills measured in kindergarten were predictive of reading skills in 5th grade. The composite score of ToM skills measured in 5th grade was also positively correlated with math and reading skills at both time points. The sizes of these correlations were small to medium (see Table 4). Taken separately, both SET and RME scores were significantly correlated with math scores in 5th grade (SET: r = .32, p = .002; RME: r = .27, p = .01) and reading scores in 5th grade (SET: r = .25, p = .02; RME: r = .35, p < .001). SET scores were also significantly correlated with math scores in kindergarten (r = .35, p < .001) and tended to be associated with reading scores in kindergarten (r = .18, p = .098). RME scores tended to be correlated with math scores in kindergarten (r = .20, p = .056), though no association was observed with reading scores in kindergarten (r = .16, p = .125). Therefore, overall, both measures of empathy and emotion recognition were positively associated with academic outcomes in kindergarten and fifth grade, supporting the use of a composite score of ToM in 5th grade. Interestingly, there was no correlation between ToM skills measured in kindergarten and 5th grade, suggesting that these tests may measure different constructs (see Discussion).

Finally, for exploratory purpose, we report in Table S4 the partial correlations between ToM scores measured in both kindergarten and 5th grade and academic outcomes, controlling for all executive functions (working memory, planning and self-regulation). Although the associations between ToM in 5th grade and academic achievement remained positive, they were no longer significant. This might suggest overlapping variance between ToM and executive function measures in predicting academic outcomes. However, these findings need to be interpreted with caution for two reasons. First, there is theoretical and empirical evidence suggesting that ToM and executive functions are developmentally intertwined constructs [38,101,137], with executive functions potentially supporting the emergence and expression of ToM abilities. Therefore, attempting to statistically isolate their unique contributions may create an artificial distinction that does not reflect their integrated development and may remove meaningful shared variance that is potentially important for understanding academic learning.

Because only the composite measure of ToM skills measured in 5th grade was significantly related to academic skills in both kindergarten and 5th grade, we tested whether those 5th grade ToM skills mediated the relation (1) between early and later math skills, (2) between early and later reading skills, (3) between early math skills and later reading skills, and (4) between early reading skills and later math skills (see **Fig 2**). We found that ToM skills partially mediated all relations (between early and later math skills: indirect effect = 0.087, 95% CI = [0.01, 0.18], indirect/total effect = 0.18; between early reading skills and later reading skills: indirect effect = 0.069, 95% CI = [0.007, 0.14], indirect/total effect = 0.23; between early math skills and later reading skills: indirect effect = 0.093, 95% CI = [0.02, 0.20], indirect/total effect = 0.27; between early reading skills and later math skills: indirect effect = 0.073, 95% CI = [0.007, 0.17], indirect/total effect = 0.20). Therefore, our results indicate that ToM may explain within- and cross-domain relations between early and later academic skills.

## Discussion

The present study aimed to both replicate the relation between early and later academic skills in a sample of French children followed longitudinally from kindergarten to 5th grade, but also to shed some light on the mechanisms that may underlie that relation. Our findings largely replicate the prior evidence that there are within- and cross-domain relations between early and later reading and math skills, with associations of comparable magnitude to those observed in studies from the United States (e.g., [6,7]). Therefore, although we initially hypothesized that the more uniform early education

system in France might attenuate these relations, our data do not support this conclusion (see consistent data in Norway [68] and Finland [138]). This might be because French preschool education struggles to equalize opportunities and might not provide high quality early education to all children [139], which may exacerbate individual differences in skill-building. Alternatively, it could indicate that the relation between early and later academic skills is driven by environmental factors (e.g., home learning environments [140] and genetic factors [141]) that are relatively independent from early education. Nonetheless, our study provides empirical evidence that domain-general factors such as self-regulation, working memory, and ToM partially mediate the relation between early and later academic achievement, suggesting that multiple developmental pathways likely contribute to academic continuity over time.

## Self-regulation and working memory mediate the relation between early math skills and later math skills

Our study is not the first to test whether executive functions may mediate the relation between early and later academic achievement [68,118]. However, prior studies have yielded inconsistent results. On the one hand, Watts et al. (2015) failed to find any mediating role for working memory, planning and problem solving, and attentional control in the relation between first-grade and high-school mathematics achievement. On the other hand, ten Braak et al. (2022) did show that self-regulation mediates the relation between kindergarten and 5th grade math achievement. Focusing on the exact same developmental period as ten Braak et al. (2022) and using the exact same self-regulation measure (HTKS measured in kindergarten), our findings replicate their results, as we also found that behavioral self-regulation mediated the association

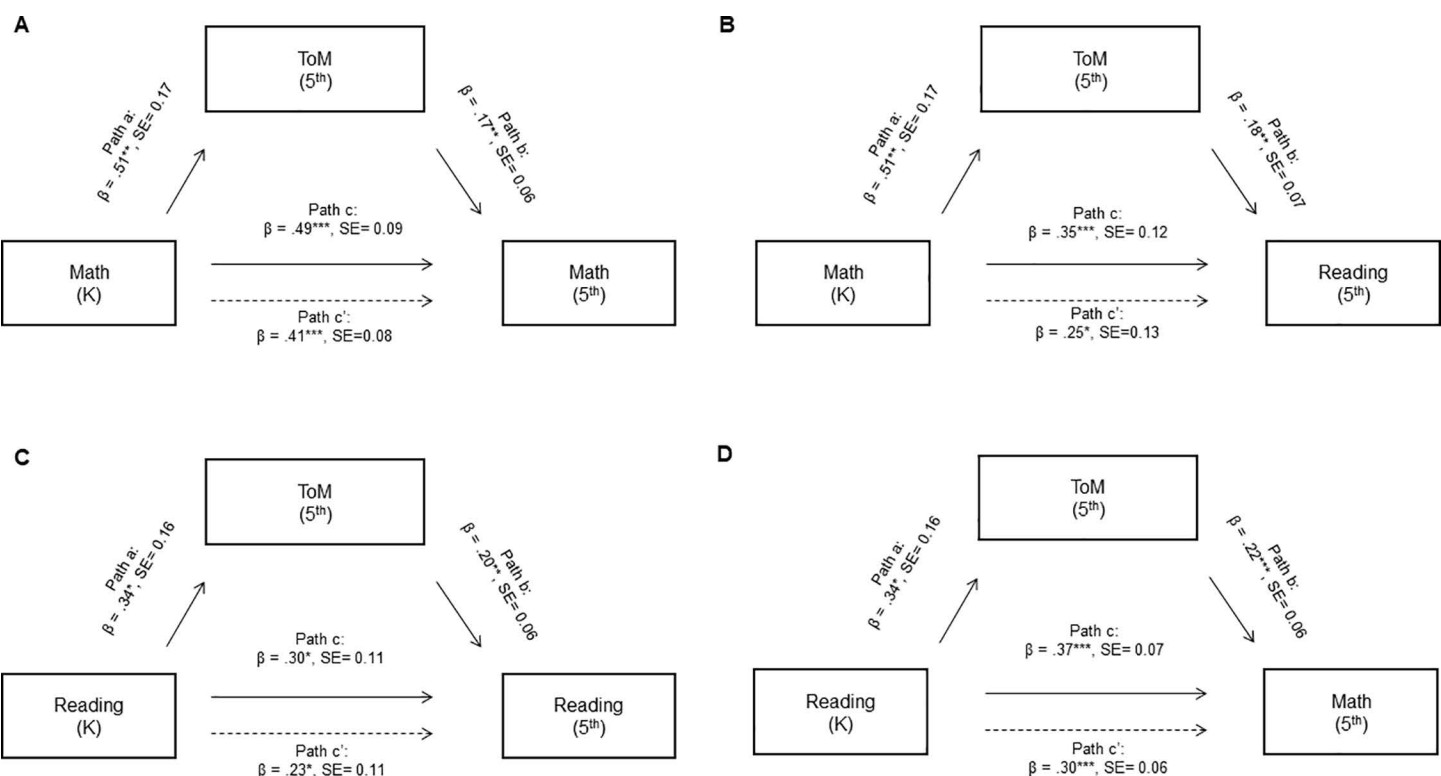

**Fig 2. Mediation Paths for Models of ToM (5th grade) Explaining the Relation Between Early and Later Academic Skills.** (A) Mediation of the relation between early and later math skills by ToM. (B) Mediation of the relation between early math and later reading skills by ToM. (C) Mediation of the relation between early and later reading skills by ToM. (D) Mediation of the relation between early reading and later math skills by ToM. Solid lines between the independent and the dependent variable are total effects, and dashed lines are direct effects. * p<.05, ** p<.01, *** p<.001.

between early and later math skills. Therefore, it is possible that behavioral self-regulation may specifically contribute to the relation between school-entry and elementary-school math (more so than high-school math), perhaps because it is particularly needed in elementary school to adapt to a relatively novel classroom environment where tasks may require sustained attention.

Our findings, however, go further than those of ten Braak et al. (2022). Indeed, we also showed that working memory was a mediator of the relation between early and later math skills. Critically, this was not the case of short-term memory, despite the fact that short-term memory skills were also associated with both early and later math skills. This suggests that the ability to merely retain information is not sufficient for explaining the relation between early and later math achievement. Rather, the active manipulation of information over a short period of time—a hallmark of working memory [142]—is the critical factor explaining the relation between early and late math achievement. This supports the idea that there is a bidirectional relation between math and working memory skills. This bidirectional relation is not only suggested by longitudinal evidence demonstrating that working memory may predict math skills and vice versa [47], but also by intervention studies showing that improving math skills enhance working memory and vice versa [52,53,70]. Therefore, early math skills may enhance working memory by providing children with opportunities to engage in tasks that require the active manipulation and organization of numerical information, such as solving arithmetic problems or recognizing patterns. In turn, enhanced working memory can further improve later math skills by enabling children to hold and process multiple steps in complex problem-solving, such as performing multi-digit calculations or understanding more abstract mathematical concepts (e.g., fractions, algebra).

Overall, our findings suggest that executive functions contribute to the observed association between early and later math achievement, consistent with developmental models that view math learning as a dynamic interaction between domain-specific and domain-general skills [143,144]. Although our findings are correlational, they support the idea that strong math skills in kindergarten may enhance executive functions, which, in turn, could foster greater academic success. This dynamic may create a positive feedback loop, where children with strong early math skills gain more opportunities to develop their executive functions, further promoting math growth. Conversely, children with weaker math skills may not encounter these same opportunities and may experience a negative feedback loop, limiting their academic development.

### ToM mediates the relation between early and later academic skills

A novel finding from the present study is that ToM skills mediated the relation between early and later math, but also the relation between early and later reading (in contrast to executive functions). This indicates a role for social cognition in academic learning more generally. Several studies suggest that higher ToM skills can help children communicate more effectively and collaborate more successfully with teachers, which in turn may support their academic achievement [81,106,145]. Additionally, strong ToM skills enhance peer relationships, a key factor in academic success [146–149]. ToM may also contribute to the development of metacognitive abilities, allowing children with enhanced ToM to apply more effective learning strategies due to an awareness of their own limitations [106,150,151]. Furthermore, the ability to understand and interpret emotional cues can improve classroom attention, further boosting academic performance [152,153]. Importantly, the relation between ToM and academic skills is likely reciprocal. While higher ToM skills may positively influence academic outcomes, developing academic skills may also foster ToM. For instance, engaging with academic tasks that require perspective-taking or understanding others' thoughts and emotions may promote ToM, creating a reinforcing cycle of learning and development. To our knowledge, our study is the first to demonstrate that social cognition is a mediator of the relation between early and later academic achievement, suggesting that future research should pay attention to social as well as executive factors when exploring the determinants of academic growth.

### Both executive functions and ToM mediate cross-domain relations between reading and math

We also found that both executive functions and ToM mediated the cross-domain relations between early and later academic achievement. Though this is a novel result for ToM, it is consistent with ten Braak et al. (2022) who also found

that self-regulation mediated a cross-domain association between math and reading. A number of previous studies have found cross-domain relations between early and later literacy and numeracy skills [6,7,10], notably supporting the idea that early math skills may critically contribute to later reading skills and vice versa (though the relation is typically stronger for math to reading than the other way around; [7]). For example, a seminal meta-analysis by Duncan et al. (2007) found that later reading skills are predicted by early math skills as much as by early reading skills. Therefore, it has been argued that early math interventions may be beneficial not only for improving math skills in the long run but also literacy skills, an idea supported by some experimental evidence [154].Yet, the mechanisms underlying the cross-domain relation between math and reading remain debated. For example, it has been proposed that the combination of conceptual and procedural competences at the heart of math learning may promote learning in the reading domain as well [7]. A number of more recent reports, however, suggest that early math skills may instead act as a proxy measure for other skills that may be more directly related to reading, including mathematical language [155], phonological awareness and symbolic processing [156,157], working memory [157], and self-regulation [68]. The present findings add novel evidence to the latter hypothesis that the cross-domain relation between math and reading is relatively non-specific and explained by domain-general skills such as executive functions and social cognition.

Interestingly, we assessed ToM both in kindergarten and in 5th grade, yet only the 5th-grade measure mediated the relation between all within- and cross-domains relations. ToM measured in kindergarten was neither associated with early math and reading skills nor with later math skills. This pattern likely reflects the developmental trajectory of ToM abilities. Indeed, children's ToM capabilities develop significantly during the preschool years [158], but continue to evolve throughout elementary school [159,160]. By age 4, most children have acquired basic first-order ToM skills, such as understanding that someone can hold a false belief about the world [158] or attributing basic intentions to others [161]. Even very young children can discriminate basic emotions on faces [162]. However, during development, children progressively acquire more advanced ToM capacities, including second-order and higher-order false beliefs [160], increasingly sophisticated mental state and emotion recognition [128], and interpretation of complex social situations [163]. Previous studies have found relatively weak associations between early basic ToM skills and later advanced ToM abilities [79,148,164], as well as modest correlations between different advanced ToM tasks [101,165]. This developmental progression may explain the lack of correlation between our kindergarten and 5th-grade ToM measures, suggesting that they assess qualitatively different constructs at different developmental stages. Specifically, the Wellman & Liu task used in kindergarten measures relatively 'basic' ToM skills, requiring children to compare their own mental states with those of others—a self-other distinction that is critical in early childhood [166]. This widely-used measure was selected as it appropriately captures ToM capabilities at the preschool stage. By 5th grade, most children may have mastered these first-order skills, necessitating more sophisticated measures. Our 5th-grade assessments—the Reading the Mind in the Eyes test [128,129] and the Story-based Empathy Task [127]—evaluate advanced ToM abilities, including attribution of complex mental states and emotions through the interpretation of social situations. These are likely more directly relevant to communication, collaboration, and emotional regulation in the classroom [76], potentially explaining their stronger mediating role.

Finally, we did not find any evidence that social behaviors such as sharing, distributing resources or solving social problems, mediate the relationship between early and later academic skills. One possible explanation for this is that these particular social factors may become more relevant to academic achievement beyond the elementary school years. As children progress through school, the social demands of the classroom and peer interactions may become increasingly complex, potentially making these behaviors more influential in middle and high school where they might contribute to collaborative learning and academic well-being. Alternatively, it is also possible that these social factors do not play a significant role in mediating the relation between early and later academic skills, suggesting that other variables are more critical in this process. Still, our findings support the notion that social cognition skills, such as ToM, may be distinct from social behaviors.

## Mediations of the relation between early and later academic skills are only partial

An interesting aspect of our results is that executive functions and ToM only partially mediated the relationship between early and later academic skills. In other words, early math and reading skills predicted later math skills even after considering individual differences in executive functions. Similarly, early math and reading skills predicted later math and reading skills even after controlling for individual differences in ToM. This indicates that early math and reading skills remain a particularly strong predictor of later math and reading skills, beyond the influence of individual differences in executive functions and social cognition. Overall, this finding aligns well with the skill-building hypothesis, which suggests that domains such as reading and math are highly cumulative, with early competencies serving as a foundation for later ones [12]. Therefore, while our results suggest that skill-building may not be the sole factor contributing to the relationship between early and later academic achievement, it likely remains a significant one.

## Limitations

It is important to acknowledge several limitations of the present study. First, the sample size was constrained by the cohort of children tested in kindergarten and by attrition between kindergarten and 5th grade. Although our final sample size may seem modest compared to some other studies [7,8,68,138], it allowed us to collect a much more comprehensive range of measures for each child, with individual assessments conducted by an experimenter. This approach contrasts with studies that typically gather a smaller range of measures, sometimes collected by teachers rather than experimenters [68,167,168], from larger samples. Our comprehensive assessment enabled us to identify which specific skills mediate the relationship between early and later academic skills. Another strength of our approach that may compensate the relatively limited sample size is the use of consistent instruments to assess reading and math skills in both kindergarten and 5th grade (e.g., the exact same test was used to measure math skills) and the evaluation of multiple skills at both time points. This consistency is rarely seen in studies, which typically employ different measures of academic skills at different time points and often examine potentially mediating skills at a single time point [81,118,169]. We believe our study provides an important first step towards evaluating these relationships in a larger sample with fewer measurements.

Second, we decided to use composite scores for certain measures across time points. This carries both advantages and limitations. While this approach allows us to capture more reliable estimates of relatively stable traits (which were significantly correlated across time points), it might obscure potential developmental changes in these capacities. Future research with larger samples might explore more complex models that separate these constructs at different time points while still examining their mediating roles.

Third, although the relationship between early and later academic skills is longitudinal, measurements of executive functions, ToM, and social behavior were collected simultaneously with academic skills in kindergarten and 5th grade. Therefore, the relations between our potential mediators and academic skills are often (though not always) cross-sectional. Furthermore, our study is correlational, and our findings do not establish the causal aspects of the mediation models. It is important to emphasize that our mediation analyses were designed to test specific pre-registered hypotheses about whether domain-general factors help explain the relationship between early and later academic skills, not to establish exclusive causal pathways. While the observed relations are consistent with our model, alternative models could also explain our results. For example, it is possible that executive functions and ToM causally influence early academic skills, which in turn may influence later academic skills.

Truly disentangling bidirectional relationships between cognitive functions and academic skills presents significant methodological challenges, particularly with observational data collected at only two time points. While alternative analytical approaches such as cross-lagged panel models could potentially provide additional insights into these bidirectional relationships, such models typically benefit from at least three measurement time points for optimal estimation of developmental trajectories [170,171] and would be statistically challenging given our sample size [172]. In a mediation framework such as the one followed here, the choice of the most plausible mediation model ultimately depends on prior

findings, logical reasoning, and theoretical considerations [173]. Our approach follows methodological precedents in this literature (e.g., [68,118]) while acknowledging that the observed relations likely reflect complex, reciprocal developmental processes.

Importantly, our model is grounded in a general theoretical framework that has guided several studies since the seminal meta-analysis by Duncan et al. (2007) [21,68,118]. Nevertheless, intervention studies are needed to substantiate causal claims, and our findings should be viewed as an initial critical test of the models discussed. Future studies with larger samples and additional measurement time points could employ alternative analytical approaches to more comprehensively examine potential bidirectional relationships between executive functions, theory of mind, and academic skills across development.

## Conclusion

In sum, our study contributes to the growing body of research exploring the relation between early and later academic achievement. Our findings confirm that early math and reading skills are strong predictors of both later math and reading skills, even in a context of relatively homogenous early education. Moreover, we demonstrate that domain-general factors, such as executive functions and ToM, play a significant, though partial, mediating role in these relationships. These results suggest that while skill-building remains an important factor in academic development, it interacts dynamically with cognitive and social-cognitive processes. Future interventional research is necessary to establish causality and to improve our understanding of how early academic skills and cognitive factors may shape long-term academic outcomes.

## Supporting information

**Table S1. Descriptions of the tests measuring other literacy and numeracy skills by grade.**
(DOCX)

**Table S2. Descriptive statistics (raw scores).**
(DOCX)

**Table S3. Sensitivity analyses using separate time point measures for working and short-term memory.**
(DOCX)

**Table S4. Partial correlations between ToM scores measured at both kindergarten and 5th grade and academic outcomes.**
(DOCX)

## Acknowledgments

We thank Camille Roullet, Lucy-Jane Durand, and Laurine Milon for their assistance in collecting the data. We are also grateful to all of the children and teachers who participated in this study.

## Author contributions

**Conceptualization:** Sarah Le Diagon, Marie Jacquel, Jean-Baptiste Van der Henst, Jérôme Prado.

**Data curation:** Sarah Le Diagon.

**Formal analysis:** Sarah Le Diagon.

**Funding acquisition:** Jérôme Prado.

**Investigation:** Sarah Le Diagon, Marie Jacquel.

**Methodology:** Sarah Le Diagon, Marie Jacquel, Jean-Baptiste Van der Henst, Jérôme Prado.

**Project administration:** Sarah Le Diagon.

**Supervision:** Jean-Baptiste Van der Henst, Jérôme Prado.

**Visualization:** Sarah Le Diagon, Jérôme Prado.

**Writing – original draft:** Sarah Le Diagon, Jean-Baptiste Van der Henst, Jérôme Prado.

**Writing – review & editing:** Jean-Baptiste Van der Henst, Jérôme Prado.

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
