## [Decision Letter · Decision Letter 0]

20 Dec 2024

PONE-D-24-50558Individual differences in executive functions and theory of mind mediate the relation between academic skills from kindergarten to 5th gradePLOS ONE

Dear Dr. Le Diagon,

Thank you for submitting your manuscript to PLOS ONE. After careful consideration, we feel that it has merit but does not fully meet PLOS ONE’s publication criteria as it currently stands. Therefore, we invite you to submit a revised version of the manuscript that addresses the points raised during the review process.

We look forward to receiving your revised manuscript.

Kind regards,

Jie Wang, Ph.D.

Academic Editor

PLOS ONE

2. We noted in your submission details that a portion of your manuscript may have been presented or published elsewhere. [Yes. A paper using the same dataset as the one we are submitting to you has been submitted to another journal and is currently under review (citation of the paper: Le Diagon, S., Van der Henst, J. B., & Prado, J. (submitted). Emergence and fadeout of effects from an early childhood Montessori intervention: A longitudinal randomized controlled study from preschool to 5th grade). Indeed, the children we followed longitudinally from kindergarten to fifth grade differed in the type of pedagogy in kindergarten (Montessori versus Conventional). The aim of that paper was to assess whether the short-term effects (i.e., in kindergarten) previously identified for the Montessori pedagogy on children's academic, cognitive, and social skills were still present in the long term (i.e., in fifth grade) and whether other differences could be observed. Our main findings show a fade-out of the advantage in reading conferred by the Montessori pedagogy in kindergarten. We also observed a difference in math problem-solving skills in fifth grade, with an advantage for children who had followed the Montessori pedagogy in kindergarten. Since the objectives of this study differ from those of the paper we are submitting to you, there is no overlap between the two.] Please clarify whether this [conference proceeding or publication] was peer-reviewed and formally published. If this work was previously peer-reviewed and published, in the cover letter please provide the reason that this work does not constitute dual publication and should be included in the current manuscript.

Additional Editor Comments:

The authors need to address the comments raised by the three reviewers. Reviewer 2 had concerns about the rationale behind the mediation models. The authors may consider reconceptualizing the models, and justification for the changed or unchanged models should be included in the revision.

Reviewers' comments:

Reviewer's Responses to Questions

**Comments to the Author**

1. Is the manuscript technically sound, and do the data support the conclusions?

Reviewer #1: Yes

Reviewer #2: Partly

Reviewer #3: Yes

2. Has the statistical analysis been performed appropriately and rigorously? 

Reviewer #1: Yes

Reviewer #2: Yes

Reviewer #3: Yes

3. Have the authors made all data underlying the findings in their manuscript fully available?

Reviewer #1: Yes

Reviewer #2: Yes

Reviewer #3: Yes

4. Is the manuscript presented in an intelligible fashion and written in standard English?

Reviewer #1: Yes

Reviewer #2: Yes

Reviewer #3: Yes

5. Review Comments to the Author

Reviewer #1: This study investigates the predictive value of early academic skills on later academic outcomes in a French sample, exploring whether this association can be partially explained by domain-general mechanisms—specifically, executive functions (EF) and ToM. The authors find that early math and reading abilities predict later academic success both within and across domains, aligning with previous findings primarily from U.S. samples. They also show that self-regulation, working memory, and ToM mediate these relations.

The study addresses a relevant gap by examining these associations in a French context with more homogeneous early education; however, several points warrant clarification or further discussion:

(1) The authors should discuss ToM development more explicitly. Notably, the period between kindergarten and fifth grade encompasses critical advancements in children’s understanding of mental states. Differentiating between basic (first-order) ToM skills and more advanced (higher-order) ToM understanding would help contextualize the chosen measures and interpret the stability (or lack thereof) of ToM abilities.

(2) On page 6, the authors suggest that the social nature of schools is often overlooked in studies of academic achievement. However, there is a substantial body of research linking ToM to various academic domains (e.g., reading, math, scientific reasoning) and investigating these associations longitudinally (e.g., Lecce et al., 2021; Kloo et al., 2022; Osterhaus et al., 2024; see also Tompkins et al., 2024, for a meta-analysis). This literature should be included in the literature review.

(3) Additional information about the ToM assessments, particularly at fifth grade, would be helpful. The correlation between the two ToM tasks (r = .30) is modest, and the absence of a longitudinal correlation between ToM measures from kindergarten to fifth grade is unexpected. This might reflect conceptual differences between the measures (e.g., RMET and SET). I wonder if the decision to consider a single ToM score is really justified. Reporting separate correlations between academic achievement and each ToM task would clarify how each measure relates to academic outcomes.

(4) As ToM is related to EF, it would be informative to report partial correlations between ToM and academic achievement controlling for EF.

Reviewer #2: This manuscript (MS) examines the mediating roles of executive functions (EF) and social cognition and behavior (SCB) in the longitudinal associations between reading and math at kindergarten and 5th grade in 95 children. The MS is generally well-written, in terms of linguistic clarity. The topic is of interest as there are limited studies examining longitudinal pathways of development in diverse settings. However, I have some concerns with the MS in its current state. In particular, conceptual clarity needs to be enhanced in several areas.

1. First, it is unclear and rather perplexing why data from two timepoints several years apart are combined in a composite score. What do the scores represent? How do they fit in a model where, e.g., math scores from the same test at kindergarten and Grade 5 are separate manifest variables? The underlying logic needs to be explicated.

2. The rationale and justification for the mediation models tested should be articulated. Early EF/SCB mediating the link between early to later academic skills is problematic as the early EF/SCB is measured at the same timepoint as the early academic skills. There is much evidence of EF/SCB contributing to academic skills and, as the authors pointed out in their literature reviews, studies have found earlier academic skills predicting later EF/SCB. The current analyses and findings do not rule out, theoretically and empirically, the alternative model where early academic skills mediate the link between early EF/SCB and later academic skills. It is unclear what is being examined in models with the K/5th composite measures. The authors highlighted literature suggesting bidirectional associations amongst the constructs but did not account for them in their models. The piecemeal approach ignores what might be extensive overlapping variance among the early and late EF, SCB, math, and reading skills.

3. It is strongly recommended that the authors reconsider the appropriate analytical model that will address the research gaps/questions in a more comprehensive and conclusive manner, such as a cross-lag model that considers concurrent and longitudinal autoregressive and cross-lag associations amongst EF, SCB, reading and math. There may be power limitations with the small sample size; the authors may want to sharpen their focus. As the authors highlighted, a key strength of the study lies in its multiple timepoints and measures. However, the current analysis of this valuable dataset falls short of realizing its full potential.

Reviewer #3: This paper reports on a longitudinal study on the development of reading and math from kindergarten to fifth grade, investigating cross-domain associations and mediation affects of domain-general cognitive skills executive functions and theory of mind. It's commendable that the authors pre-registered their analyses and have shared their data. The paper is clearly written and the results and conclusions are straightforward and consistent with prior research. I have only two minor comments. First, the figure resolution is very poor in this submitted version and should be addressed. Second, the hypothesis of universal pre-k potentially reducing variability in kindergarten math performance is a very interesting and important question, but not sufficiently engaged with in the current paper. The authors rightfully note in the discussion section that the pattern of results they found could either be explained by inequalities in access to high quality early education (or in other words failure of policy implementation), or cognitive factors or other difference that cannot be changed by school (what could those be? genetic influences?). Research on the debates of the long run impacts of early education are discussed in the paper (e.g. Burchinal et al., 2024), but it's not clear how the current paper contributes to this debate given the findings are inconclusive. What would be needed to tease apart these competing explanations? What are the future research directions? While I acknowledge the need to replicate findings in more diverse school systems and culture, I am not convinced these findings really have anything to offer to this particular debate and the way it is currently laid out is very speculative. The authors may be better off focusing on the more novel finding that ToM emerged as a significant predictor.

6. PLOS authors have the option to publish the peer review history of their article (what does this mean? ). If published, this will include your full peer review and any attached files.

**Do you want your identity to be public for this peer review?** For information about this choice, including consent withdrawal, please see our Privacy Policy .

Reviewer #1: No

Reviewer #2: No

Reviewer #3: No

---

## [Author Response · Author response to Decision Letter 1]

14 Mar 2025

Response to reviews

We wish to thank the reviewers for their comments on our manuscript. Below, we indicate the responses to the specific issues raised and the corresponding changes that we have made in the revised manuscript.

Reviewer #1

This study investigates the predictive value of early academic skills on later academic outcomes in a French sample, exploring whether this association can be partially explained by domain-general mechanisms—specifically, executive functions (EF) and ToM. The authors find that early math and reading abilities predict later academic success both within and across domains, aligning with previous findings primarily from U.S. samples. They also show that self-regulation, working memory, and ToM mediate these relations.

The study addresses a relevant gap by examining these associations in a French context with more homogeneous early education; however, several points warrant clarification or further discussion:

Response: We thank the reviewer for this positive assessment of our paper.

(1) The authors should discuss ToM development more explicitly. Notably, the period between kindergarten and fifth grade encompasses critical advancements in children’s understanding of mental states. Differentiating between basic (first-order) ToM skills and more advanced (higher-order) ToM understanding would help contextualize the chosen measures and interpret the stability (or lack thereof) of ToM abilities.

Response: We have followed the reviewer’s suggestion and have now significantly expanded the relevant paragraph in the discussion to describe the development of ToM skills. On p. 28, we now state:

“Interestingly, we assessed ToM both in kindergarten and in 5th grade, yet only the 5th-grade measure mediated the relation between all within- and cross-domains relations. ToM measured in kindergarten was neither associated with early math and reading skills nor with later math skills. This pattern likely reflects the developmental trajectory of ToM abilities. Indeed, children's ToM capabilities develop significantly during the preschool years [157], but continue to evolve throughout elementary school [158,159]. By age 4, most children have acquired basic first-order ToM skills, such as understanding that someone can hold a false belief about the world [157] or attributing basic intentions to others [160]. Even very young children can discriminate basic emotions on faces [161]. However, during development, children progressively acquire more advanced ToM capacities, including second-order and higher-order false beliefs [159], increasingly sophisticated mental state and emotion recognition [127], and interpretation of complex social situations [162]. Previous studies have found relatively weak associations between early basic ToM skills and later advanced ToM abilities [79,147,163], as well as modest correlations between different advanced ToM tasks [101,164]. This developmental progression may explain the lack of correlation between our kindergarten and 5th-grade ToM measures, suggesting that they assess qualitatively different constructs at different developmental stages. Specifically, the Wellman & Liu task used in kindergarten measures relatively ‘basic’ ToM skills, requiring children to compare their own mental states with those of others—a self-other distinction that is critical in early childhood [165]. This widely-used measure was selected as it appropriately captures ToM capabilities at the preschool stage. By 5th grade, most children may have mastered these first-order skills, necessitating more sophisticated measures. Our 5th-grade assessments—the Reading the Mind in the Eyes test [127,128] and the Story-based Empathy Task [126]—evaluate advanced ToM abilities, including attribution of complex mental states and emotions through the interpretation of social situations. These are likely more directly relevant to communication, collaboration, and emotional regulation in the classroom [76], potentially explaining their stronger mediating role.”

(2) On page 6, the authors suggest that the social nature of schools is often overlooked in studies of academic achievement. However, there is a substantial body of research linking ToM to various academic domains (e.g., reading, math, scientific reasoning) and investigating these associations longitudinally (e.g., Lecce et al., 2021; Kloo et al., 2022; Osterhaus et al., 2024; see also Tompkins et al., 2024, for a meta-analysis). This literature should be included in the literature review.

Response: We agree that this statement was unwarranted and have now removed it from the manuscript. On p. 6, we have now included all of the studies pointed out by the reviewer. This revised paragraph reads:

“Critically, schools are also social environments. As such, learning at school is fundamentally a social process [73–77] that involves managing relationships with teachers and peers. Therefore, there may be a relation between academic skills and children’s abilities to navigate the social world. These abilities may encompass social cognition skills, which center on the ability to understand our own and others' mental states (e.g., thoughts, intentions, emotions, desires, or beliefs) but also include the ability to perceive social relationships (alliance, friendship, dominance) and form beliefs associated with social categories or gender (e.g., stereotypes). These abilities may also include social behaviors, which describe the behaviors that are needed to develop socially appropriate responses and build positive relationships with others (e.g., sharing, communicating, cooperating, and knowing how to solve social problems). The idea that academic achievement relates to social cognition and behavior is supported by a number of cross-sectional and short-term longitudinal studies. For example, proficiency in Theory of Mind (ToM)—the ability to infer mental states—is linked to greater academic skills (see [78] for a meta-analysis) both in preschool [38,79,80] and in elementary school [81–85]. Other studies have found a relation between children's socio-emotional skills (e.g., emotion recognition and emotion understanding) and math and reading skills in preschool [86–89] and in elementary school [90–92]. Social skills (e.g., cooperating, communicating, and sharing with others, being empathetic, taking responsibility, accepting others) have also been found to be associated with concurrent levels of academic achievement [93–96]”

(3) Additional information about the ToM assessments, particularly at fifth grade, would be helpful. The correlation between the two ToM tasks (r = .30) is modest, and the absence of a longitudinal correlation between ToM measures from kindergarten to fifth grade is unexpected. This might reflect conceptual differences between the measures (e.g., RMET and SET). I wonder if the decision to consider a single ToM score is really justified. Reporting separate correlations between academic achievement and each ToM task would clarify how each measure relates to academic outcomes.

Response: We thank the reviewer for this comment. We have added in the manuscript an analysis of correlations between academic outcomes and SET and RME taken separately to provide a more detailed picture of associations. On p. 15, we have added a sentence indicating that we will present correlations considering both the composite measure of ToM and each individual measure. This sentence reads:

“However, we also report correlations between each measure taken separately and academic outcomes to assess the specificity of measures of empathy and emotion recognition.”

On p. 22, we now report these correlations. Overall, we find significant or near-significant associations between academic outcomes and measures of SET and RME taken separately (with the exception of RME and reading score in kindergarten), which in our opinion still supports the use of a more comprehensive composite score. This edited paragraph now reads:

“Finally, we examined whether ToM and social behavior mediated the relation between early and later reading and math skills. We did not find that social behavior was associated with either math or reading skills in both kindergarten and 5th grade. However, ToM skills measured in kindergarten were predictive of reading skills in 5th grade. The composite score of ToM skills measured in 5th grade was also positively correlated with math and reading skills at both time points. The sizes of these correlations were small to medium (see Table 4). Taken separately, both SET and RME scores were significantly correlated with math scores in 5th grade (SET: r=.32, p=.002; RME: r=.27, p=.01) and reading scores in 5th grade (SET: r=.25, p=.02; RME: r=.35, p<.001). SET scores were also significantly correlated with math scores in kindergarten (r=.35, p<.001) and tended to be associated with reading scores in kindergarten (r=.18, p=.098). RME scores tended to be correlated with math scores in kindergarten (r=.20, p=.056), though no association was observed with reading scores in kindergarten (r=.16, p=.125). Therefore, overall, both measures of empathy and emotion recognition were positively associated with academic outcomes in kindergarten and first grade, supporting the use of a composite score of ToM in 5th grade. Interestingly, there was no correlation between ToM skills measured in kindergarten and 5th grade, suggesting that these tests may measure different constructs (see Discussion).”

(4) As ToM is related to EF, it would be informative to report partial correlations between ToM and academic achievement controlling for EF.

Response: We have now added in the manuscript a supplementary table (Table S4) that includes these partial correlations. Note that, although the correlations between ToM skills in 5th grades and academic outcomes remained positive, they no longer reached significance. While this might suggest overlapping variance between ToM and EF measures in predicting academic outcomes, we interpret these findings cautiously for several reasons.

First, there is substantial theoretical and empirical evidence suggesting that ToM and EF are developmentally intertwined constructs (Blair & Razza, 2007; Devine et al., 2016; Carlson et al., 2013), with executive functions potentially supporting the emergence and expression of ToM abilities. Therefore, attempting to statistically isolate their unique contributions may create an artificial distinction that does not reflect their integrated development. Second, the statistical control of EF when examining the relation between ToM and academic achievement potentially removes meaningful shared variance that is potentially important for understanding academic learning.

On p. 23, we have added a paragraph about these results. It reads:

“Finally, for exploratory purpose, we report in Table S4 the partial correlations between ToM scores measured in both kindergarten and 5th grade and academic outcomes, controlling for all executive functions (working memory, planning and self-regulation). Although the associations between ToM in 5th grade and academic achievement remained positive, they were no longer significant. This might suggest overlapping variance between ToM and executive function measures in predicting academic outcomes. However, these findings need to be interpreted with caution for two reasons. First, there is theoretical and empirical evidence suggesting that ToM and executive functions are developmentally intertwined constructs [38,101,136], with executive functions potentially supporting the emergence and expression of ToM abilities. Therefore, attempting to statistically isolate their unique contributions may create an artificial distinction that does not reflect their integrated development and may remove meaningful shared variance that is potentially important for understanding academic learning.”

Reviewer #2

This manuscript (MS) examines the mediating roles of executive functions (EF) and social cognition and behavior (SCB) in the longitudinal associations between reading and math at kindergarten and 5th grade in 95 children. The MS is generally well-written, in terms of linguistic clarity. The topic is of interest as there are limited studies examining longitudinal pathways of development in diverse settings. However, I have some concerns with the MS in its current state. In particular, conceptual clarity needs to be enhanced in several areas.

1. First, it is unclear and rather perplexing why data from two timepoints several years apart are combined in a composite score. What do the scores represent? How do they fit in a model where, e.g., math scores from the same test at kindergarten and Grade 5 are separate manifest variables? The underlying logic needs to be explicated.

Response: We apologize for the lack of clarity regarding these composite scores. We have revised our manuscript to provide explicit justification for this methodological choice and to clarify the rationale behind our approach.

First, we have added the following explanation in the Methods section. On p. 15, a new paragraph now reads:

“For working memory and sharing behavior, we created composite scores across time points because: (1) identical instruments were used at both time points (Corsi Block Tapping task for working memory measures and Dictator Game for sharing behavior), (2) these measures showed significant correlations between kindergarten and 5th grade assessments (r = .29, p < .005 for working memory; r = .34, p < .001 for sharing behavior), and (3) these constructs represent relatively stable individual traits that have demonstrated moderate stability across development in previous longitudinal research [123,124]. Therefore, composite scores represent the general capacity of children in working memory and sharing behavior across the studied developmental period, allowing us to examine how these traits might mediate the relation between early and later academic skills (note that we also created a composite measure of short-term memory so that it was comparable to the composite measure of working-memory). Nonetheless, we also conduct separate mediation analyses using separate time point measures rather than composites to examine the sensitivity of our findings.”

Second, in the Discussion section (p. 31), we have acknowledged the limitations of this approach (while also acknowledging its benefits):

"Second, we decided to use composite scores for certain measures across time points. This carries both advantages and limitations. While this approach allows us to capture more reliable estimates of relatively stable traits (which were significantly correlated across time points), it might obscure potential developmental changes in these capacities. Future research with larger samples might explore more complex models that separate these constructs at different time points while still examining their mediating roles."

Third, we have conducted sensitivity analyses using the separate time point measures instead of the composites to verify the robustness of our findings. These results are now reported in Supplementary Table S3 and summarized in the manuscript (p. 21):

“Because working-memory and short-term memory measures were calculated using composite scores across time points, we also conducted sensitivity analyses using separate time point measures rather than composites (see Table S3). These analyses revealed that kindergarten working memory significantly mediated the relationship between early and later math skills (indirect effect = 0.17, 95% CI [0.06, 0.33], p = 0.002), whereas 5th grade working memory showed a smaller and non-significant mediation effect (indirect effect = 0.04, 95% CI [-0.02, 0.14], p = 0.20). While the point estimates might suggest potentially stronger mediation effects for working memory measure at kindergarten, the overlapping confidence intervals indicate that we cannot conclusively determine that one time point's measurement provides a significantly stronger mediating effect than the other. Neither kindergarten nor 5th grade short-term memory significantly mediated the relationship between early and later math skills.”

2. The rationale and justifi

---

## [Decision Letter · Decision Letter 1]

28 Apr 2025

Individual differences in executive functions and theory of mind mediate the relation between academic skills from kindergarten to 5th grade

PONE-D-24-50558R1

Dear Dr. Le Diagon,

We’re pleased to inform you that your manuscript has been judged scientifically suitable for publication and will be formally accepted for publication once it meets all outstanding technical requirements.

Kind regards,

Jie Wang, Ph.D.

Academic Editor

PLOS ONE

Additional Editor Comments (optional):

Reviewers' comments:

Reviewer's Responses to Questions

**Comments to the Author**

1. If the authors have adequately addressed your comments raised in a previous round of review and you feel that this manuscript is now acceptable for publication, you may indicate that here to bypass the “Comments to the Author” section, enter your conflict of interest statement in the “Confidential to Editor” section, and submit your "Accept" recommendation.

Reviewer #3: All comments have been addressed

2. Is the manuscript technically sound, and do the data support the conclusions?

Reviewer #3: Yes

3. Has the statistical analysis been performed appropriately and rigorously? 

Reviewer #3: Yes

4. Have the authors made all data underlying the findings in their manuscript fully available?

Reviewer #3: Yes

5. Is the manuscript presented in an intelligible fashion and written in standard English?

Reviewer #3: Yes

6. Review Comments to the Author

Reviewer #3: (No Response)

7. PLOS authors have the option to publish the peer review history of their article (what does this mean? ). If published, this will include your full peer review and any attached files.

**Do you want your identity to be public for this peer review?** For information about this choice, including consent withdrawal, please see our Privacy Policy .

Reviewer #3: No

---

## [Editor Report · Acceptance letter]

PONE-D-24-50558R1

PLOS ONE

Dear Dr. Le Diagon,

I'm pleased to inform you that your manuscript has been deemed suitable for publication in PLOS ONE. Congratulations! Your manuscript is now being handed over to our production team.

Kind regards,

on behalf of

Dr. Jie Wang

Academic Editor

PLOS ONE